# Evaluation of Triclosan coated suture in obstetrical surgery: A prospective randomized controlled study (NCT05330650)

Wael Mbarki[1,2☉], Hajer Bettaieb[1,2☉]*, Nesrine Souayeh[1,2], Idriss Laabidi[1,2], Hadhemi Rouis[1,2], Soumaya Halouani[1,2], Rami Boufarghine[1,2], Maha Bouyahia[2,3], Rahma Bouhmida[2,3], Mariem Ouederni[2,3], Anissa Ben Amor[2,4], Amal Chermiti[2,4], Hadir Laamiri[2,4], Amira Lika[2,5], Imen Chaibi[2,5], Hedhili Oueslati[1,2], Najeh Hsayaoui[1,2], Chaouki Mbarki[1,2☉]

1 Department of Gynecology and Obstetrics, Ben Arous Hospital, Ben Arous, Tunisia, 2 Faculty of Medicine of Tunis, University of Tunis El Manar, Tunis, Tunisia, 3 Department of Gynecology and Obstetrics, Aziza Othmena Hospital, Tunis, Tunisia, 4 Department of Gynecology and Obstetrics, Mongi Slim Hospital, La Marsa, Tunisia, 5 Department of Anesthesia, Ben Arous Hospital, Ben Arous, Tunisia

☉ These authors contributed equally to this work.
* hajer.bettaieb@fmt.utm.tn

**Data Availability Statement:** The data underlying the results presented in the study are available from Data Set: 10.6084/m9.figshare.21330624.

## Abstract

### Objectives

To assess the effectiveness of Triclosan coated suture in reducing surgical site infections (SSIs) rate after caesarian delivery (CD).

### Study design

Three hundred eighty patients were randomly assigned to closure with polyglactin non coated suture VICRYL, or with polyglactin coated suture VICRYL Plus after caesarian section. The primary outcome was the rate of SSIs within 30 days after surgery and secondary outcomes were the rate of wound healing complications.

### Results

SSI rate was 2.5% in Triclosan group compared to 8.1% with non-coated suture. Use of Triclosan coated suture (TCS) was associated with 69% reduction in SSI rate (p = 0.037; ORa:0.294; 95% CI:0.094–0.921). The use of Triclosan coated suture was associated with statistically lower risk of wound oedema (2.5% vs 10%), (p = 0.019; OR:0.595), dehiscence (3.8% vs 10.6%), (p = 0.023; OR:0.316) and hematoma (p = 0.035; OR:0.423).

### Conclusion

Our results confirm the effectiveness of Triclosan coated suture in reducing SSI rate and wound healing disturbances.

### Trial registration

Registered at ClinicalTrials.gov / ID (NCT05330650).

**Funding:** The authors received no specific funding for this work.

**Competing interests:** The authors have declared that no competing interests exist.

## Introduction

Surgical site infections are a common and serious complication of all surgical interventions [1]. The pooled incidence of SSI is estimated to 11.8 per 100 of surgical procedures in low-and medium-incomes countries (LMICs) [2].

SSIs results in substantial clinical and economic burden. In fact, prolonged hospital stay, increasing rate of readmission and reintervention, reduction in measures quality of life and high morality are associated with SSIs [3].

Risk factors for the development of surgical site infection can be broadly divided in patient-related and intervention-related (surgical) factors. Most identified patient-related risk factors are age, diabetes, obesity, smoking, malnutrition and immunosuppression. Since these factors are either non modifiable or difficult to control, focus has been placed on limiting surgical factors [4].

Microorganism colonization of suture materials during surgery, has been associated with a higher risk of SSI development; since bacteria can create a protective biofilm and resist to both antimicrobial treatment and to the host immune system [5].

To avoid this risk of colonization, sutures with antimicrobial activity such as Triclosan has been developed. In fact this antimicrobial agent is used to coat suture materials with a proven in vitro and in vivo efficiency on both gram-positive and gram-negative germs [6]. Effectiveness of TCS in reducing SSIs was demonstrated in several studies. A recent metanalysis including 11 957 patients across 25 randomized controlled trials (RCTs) showed that TCS was more effective in reducing SSI risk by 27 percent compared to non-coated suture [1]. Data from twelve other metanalysis concluded that TCS reduced SSI risk by 24 to 39% [3].

Despite these broad evidences, literature lacks data evaluating the effectiveness of TCS use in obstetric surgery. This data insufficiency is even more remarkable in countries with high SSIs incidence such as Tunisia.

The objectives of this study were to assess the effectiveness of Triclosan coated suture in reducing SSIs rate after caesarian delivery (CD).

## Materials and methods

### Ethics statement

The study was approved by the local ethic committee of Ben Arous hospital, Tunisia. Written consent for participation was obtained from each patient (Appendix 1). Enrollment of the first patient started on November 1st 2020. Because all protocols were originally written in French, this study was registered on ClinicalTrials.gov (NCT05330650) on August 4th 2022, after participants enrolment started, due to translation and administrative process delay. Yet registration in ClinicalTrials.gov is not mandated by Tunisian law. The full protocol is available on ClinicalTrials.gov site.

Data collection was conducted in compliance with Tunisian laws regarding personal data protection. The authors confirm that all ongoing and related trials for this procedure are registered.

### Study design and participant selection

We performed a mono-centric controlled randomized trial with patients, surgeons and outcome's assessor triple blinded to treatment. The study was conducted over eight months between November 2020 and June 2021 in the obstetrics and gynecology department of Ben Arous hospital, Tunisia. Pregnant women with elective or emergency caesarian delivery were included. Were excluded from this study all women with either suspected or confirmed

chorioamnionitis or SARS-COV2 infection. Patients' enrolment started on November 1st 2020 and follow-up of the last included patient ended on June 30th 2021.

Once written informed consent was obtained, patients eligible for the study, were randomized intraoperatively into two groups:

Group 1: Suture with polyglactin non coated suture VICRYL

Group 2: Suture with polyglactin coated suture VICRYL Plus

## Randomization and blinding

The randomization sequence was computer generated using Random Allocation Software 1.0.0 (Freeware). The patients were block randomized with equal block sizes of 20 items per block (allocation of patients per block is 1:1).

The suture materials for each group were placed in opaque sealed boxes.

The randomization sequence was afterwards provided, in the operative room, in opaque envelopes to a scrub nurse who was not involved in patient's follow-up. The nurse opened the envelope discreetly and delivered the suture material to the surgeon without uncovering the package, right before wound closure. Macroscopically, it was impossible to distinguish Vicryl® from Vicryl Plus.

## Sample size calculation

The rate of SSI after Caesar section in our center was 9% in 2019. Thus, the assumed expected wound infection rates were 2% after using coated suture versus 9% for the control group. On the basis of a two-sided chi-square test for equality of proportions, the study was expected to have 80% power to detect a relative risk reduction of 5%. A total of 296 patients were estimated to be needed, thus, 148 patients were included in each arm. The G*Power 3.1.9.6 software was used to calculate the sample size.

## Surgical procedure

Povidone iodine was used for skin preparation. A perioperative antibiotic prophylaxis was administered to all patients. Perioperative hair removal was proscribed. Pfannenstiel incision was performed in all cases. Uterus and aponeurosis closure was performed with an overlock suture with 1/0 thread. Gloves were changed after aponeurosis closure. Subcutaneous tissue was sutured with 2/0 thread. Skin was closed with intradermic suture with 2/0 thread.

## Follow up

Patients were followed 30 days after surgery, and four visits were programmed:

**Visit 1:** Two days after surgery.

**Visit 2:** 7 days after surgery.

**Visit 3:** 15 days after surgery.

**Visit 4:** 30 days after surgery.

In each visit, wound surveillance was performed by the resident in charge on the ward, and if an SSI was suspected the wound was photographed and the patient was referred to a senior surgeon to confirm the diagnosis.

## Diagnosis of SSI

We adopted the Centers for Disease Control's definition (CDC) to diagnose SSI [7]:

Superficial Incisional SSI Infection occurs within 30 days after the operation and infection involves only skin or subcutaneous tissue of the incision and at least one of the following:

1. Purulent drainage, with or without laboratory confirmation, from the superficial incision.

2. Organisms isolated from an aseptically obtained culture of fluid or tissue from the superficial incision.

3. At least one of the following signs or symptoms of infection: pain or tenderness, localized swelling, redness, or heat and superficial incision is deliberately opened by surgeon, unless incision is culture-negative.

4. Diagnosis of superficial incisional SSI by the surgeon or attending physician.

Deep Incisional SSI Infection occurs within 30 days after the operation if no implant is left in place or within 1 year if implant is in place and the infection appears to be related to the operation and infection involves deep soft tissues (eg, fascial and muscle layers) of the incision and at least one of the following:

1. Purulent drainage from the deep incision but not from the organ/space component of the surgical site.

2. A deep incision spontaneously dehisces or is deliberately opened by a surgeon when the patient has at least one of the following signs or symptoms: fever (38˚C), localized pain, or tenderness, unless site is culture-negative.

3. An abscess or other evidence of infection involving the deep incision is found on direct examination, during reoperation, or by histopathologic or radiologic examination.

4. Diagnosis of a deep incisional SSI by a surgeon or attending physician

## Outcomes

**Primary outcome.** The rate of SSIs within 30 days after surgery.
**Secondary outcomes.** The rate of wound healing complications.

## Statistical analysis

Statistical analysis was performed according to the intention-to-treat principle.

To address risk factors involved in SSI development other than suture type, a bivariate analysis using the chi-squared test for nominal variables, the independent Student's t-test for continuous variables with normal distributions, and the nonparametric Mann-Whitney U test for continuous variables with skewed distributions was first conducted on all independent variables. Any variable with a statistically significant response, or very close to it (p <0.10), was considered a potential confounding factor. A multivariate logistic regression model (Wald test), which considered the impact of all possible confounding factors identified in the bivariate analysis, was used to measure the adjusted effect of suture type on the development of SSI. Calculation 95%CI for proportions we using an online calculator available on sample-size.net.

To compare wound healing disturbances a univariate analysis using Fisher's exact test for nominal variables and student's t test for continuous variables was conducted. A p-value < 0.05 (two-sided) was considered to be significant. Statistical analyses were performed using IBM-SPSS version 24 (IBM Corporation, Armonk, New York, USA).

## Results

Three hundred fifty patients were screened for eligibility for the study and 340 patients were enrolled. One hundred seventy women (50%) were allocated in the group 1 and 170 (50%) in the group 2.

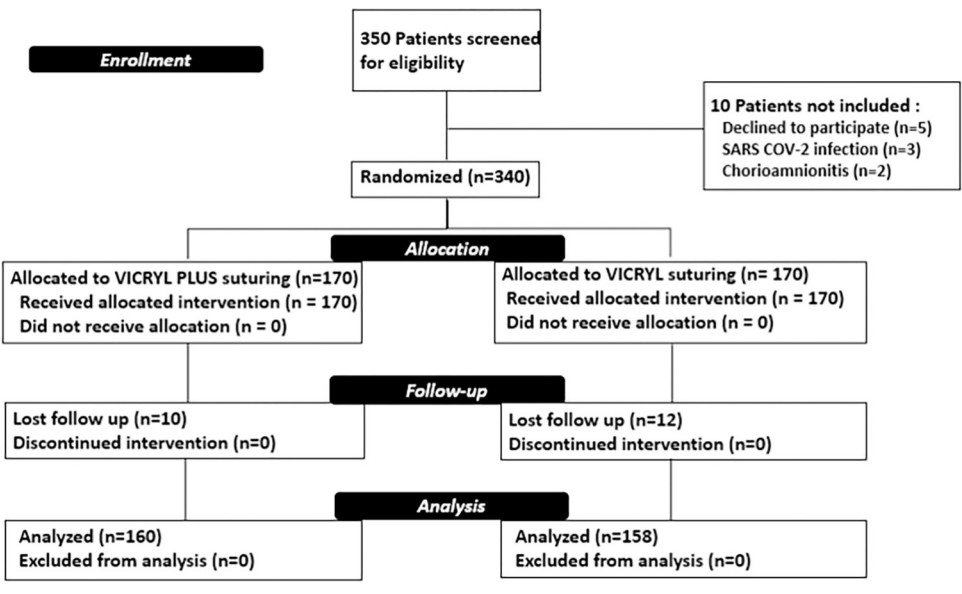

**Fig 1. CONSORT 2010 flow diagram.**

Ten women were excluded from group 1 and 12 from group 2 for an incomplete follow up. Ultimately, 160/170 (94%) patients remained in the Triclosan arm and 158/170 (93%) in the control arm (Fig 1).

The two groups were well balanced concerning patient characteristics and risk factors of SSI. The Table 1 summarizes patients characteristics and risk factors of SSI.

## Surgical site infection

The incidence of SSI in our study population was 5.3% (17/318) (95%CI [0.073–0.189]). The SSI was superficial in 15 cases and deep SSI was noted in two patients (11,7%). The two cases were managed by image guided drainage and antibiotics.

All infections were diagnosed within the first fifteen days after surgery, 35% (6/17) at the second visit and 65% (11/17) at the third visit.

When using triclosan coated suture, SSI rate was 2.5% (4/158) (95%CI [0.0008–0.049]) compared to 8.1% (13/160) (95%CI [0.044–0.134]) with non-coated suture. Non coated suture was found to increase SSI risk by 3.4 times (95%CI: 1.085–10.680). Evaluation of SSI confounding factor in bivariate analysis, found diabetes to be a confounding factor (p = 0.037; OR: 4.086; CI95%: 1.058–15.771). Bivariate analysis of SSI's risk factors is shown in the Table 2.

In multivariate regression, only use of non-coated suture was associated with a higher risk of SSI. A statistically significant reduction of 70% in SSI rate was associated with Triclosan coated suture use (ORa:0.294; 95%CI:0.094–0.921). Results of the multivariate regression are shown in Table 3.

## Wound healing disturbances

The use of Triclosan coated suture was associated with statistically lower risk of wound oedema (2.5% vs 10%), (p = 0.019; OR:0.595), dehiscence (3.8% vs 10.6%), (p = 0.023; OR:0.316) and hematoma (p = 0.035; OR:0.423).

Wound discharges were similar in the two groups (3.2% vs 5.6%) (p = 0.285). Healing time was also comparable and did not reach statistical significance (7.65 ± 1.724 vs 7.65 ± 1.724);

**Table 1. Patients' characteristics and risk of SSI.**

| | Group 1 (non-coated suture) [a] n = 160 | Group 2 (Coated suture) [a] n = 158 |
|---|---|---|
| Age (mean) | 31.94 ± 0.447 | 31.92 ± 0.443 |
| BMI (mean) | 24.10 ± 0.225 | 24.360 ± 0.241 |
| Tabacco (number) | 22 (13.8%) | 11 (7%) |
| Diabetes Miletus | 9 (5.6%) | 7 (4.4%) |
| Anemia | 37 (23.1%) | 40 (25.3%) |
| ASA score: ASA 1 ASA 2 | 124 (77.5%) 36 (22.5%) | 119 (75.3%) 32 (20.2%) |
| Parity | 2 ± 1/3 | 2 ±1/3 |
| Previous laparotomy | 90 (0.56%) | 93 (58.8%) |
| Gestational diabetes | 48 (30%) | 43 (27.2%) |
| Premature rupture of membrane > 6 H | 57 (35.6%) | 43 (27.2%) |
| Premature delivery | 23 (14.3%) | 25 (15.8%) |
| Emergency caesarian delivery | 98 (61.2%) | 71 (44.9%) |
| Rasor shaving < 48H | 49 (30.6%) | 41 (25.9%) |
| Epilation < 48H | 93 (58.1%) | 103 (68.1%) |
| Use of Drain | 7 (4.3%) | 3 (1.8%) |
| Operative time | 37.41 ± 0.409 | 38.12 ± 0.340 |

[a] Mean ± Standard error of mean (SEM) for age, body mass index (BMI), and operative time; Median ± 75th/25th percentiles for parity; percentage for tobacco, diabetes miletus, American Society of Anesthesiologists (ASA) score, previous laparotomy, gestational diabetes, premature rupture of membrane, premature delivery, emergency caesarian delivery, razor shaving, epilation and use of drain.

(p = 0.121). The Table 4 represents the results of wound healing disturbances in the two groups of the study.

## Discussion

We found an SSI risk of 2.5% in TCS coated suture group and 8.1% for the control group. The SSI rate in the control group was lower than expected. In our study, the use of TCS coated suture was found to reduce SSI rate and wound healing disturbances without increasing healing time compared with non-coated suture after caesarian section. By contrast with our assumption that coated sutures would reduce the occurrence of surgical site infection by 78%, the effective reduction in this study was 70%. It can be regarded as clinically relevant from a surgical point of view.

SSIs account for an estimated rate of 20% of all health associated infections (HAIs) globally and they are responsible of considerable morbidity and increasing health care costs [8]. To reduce this risk, several measures had been proposed. One potential strategy is to use antimicrobial coated suture. Many studies had demonstrated the effectiveness of Triclosan coated suture in reducing SSI risk. In contrast, others have failed to demonstrate protective role of TCS use. These discrepancies may be explained by differences in study population, type of procedure, or the layers where the sutures were applied [3].

In the Tristan review [9], two from the four RCTs included and evaluating Vicryl® Plus suture after abdominal surgery confirmed the superiority of Vicryl® Plus in reducing SSI rate [10–12]. In a metanalysis including 25 RCTs comparing TCS coated suture to non-coated suture, Ahmed et al had concluded to 27% reduction of SSI rate when TCS coated suture was

**Table 2. Bivariate analysis of SSI's risk factors: Chi-squared test for nominal variables, Student's t-test for continuous variables with normal distributions, Mann-Whitney U test for continuous variables with skewed distributions.**

|  | SSI<br>n = 17 | No SSI<br>n = 301 | p value | OR (95% CI) |
|---|---|---|---|---|
| Age (mean) | 32.88 ± 5.231 | 31.88 ± 5.63 | 0.473 | 1.399 (-3.757–1.746) |
| BMI (mean) | 24.782 ± 0.223 | 25.001 ±0,272 | 0.766 | 0.735 (- 1.227–1.665) |
| Tabacco (number) | 2 (11.7%) | 31 (10.2%) | 0.525 | 0.532 (0.067–4.095) |
| Diabetes Miletus | 3 (17.64%) | 13 (4.3%) | 0.041 | 4.086 (1.058–15.771) |
| Anemia | 2 (11.7%) | 75 (24.9%) | 0.218 | 0.402 (0.090–1.798) |
| ASA score = 2 | 4 (23.52%) | 64 (21.26%) | 0.812 | 1.057 (1.029–1.085) |
| Previous laparotomy | 10 (58.82%) | 173 (57.74%) | 0.913 | 1.057 (0.392–2.852) |
| Gestational diabetes | 4 (23.52%) | 87 (28.9%) | 0.786 | 0.753 (0.239–2.375) |
| Premature rupture of membrane > 6 H | 7 (41.17%) | 93 (30.9%) | 0.423 | 1.566 (0.578–4.240) |
| Premature delivery | 2 (11,7%) | 50 (15.28%) | 0.693 | 0.739 (0.164–3.341) |
| Emergency cesarian delivery | 10 (58.8%) | 159 (52.8%) | 0.804 | 1.276 (0.473–3.441) |
| Rasor shaving < 48H | 6 (35.3%) | 84 (27.9%) | 0.511 | 1.409 (0.505–3.932) |
| Epilation < 48H | 11 (64.7%) | 185 (61.4%) | 0.789 | 1.150 (0.414–3.193) |
| Use of Drain | 1 (5.89%) | 10 (3.32%) | 0.459 | 1.819 (0.219–15.097) |
| Operative time | 37.74 ± 4.818 | 37.94 ± 4.561 | 0.866 | 1.187 (-2.536–2.135) |

OR:Odds Ratio.

**Table 3. Multivariate logistic regression analysis.**

|  | p value | ORa (95% CI) |
|---|---|---|
| Diabetes | 0.101 | 0.316 (0.80–1.250) |
| Coated suture use | 0.036 | 0.294 (0.094–0.921) |

ORa: Adjusted Odds Ratio.

**Table 4. Comparison of wound healing disturbance after coated and non-coated suturing in univariate analysis.**

|  | TCS coated suture<br>n = 158 | Non coated suture<br>n = 160 | p value | OR (95% CI) |
|---|---|---|---|---|
| Oedema | 17 (3.8%) | 6 (10.6%) | 0.019 | 0.332 (0.127–0.886) |
| Dehiscence | 8 (5.1%) | 22 (13.8%) | 0.008 | 0.335 (0.144–0.776) |
| Hematoma | 9 (5.7%) | 20 (12.5%) | 0.035 | 0.423 (0.186–0.960) |
| Discharge | 5 (3.2%) | 9 (5.6%) | 0.285 | 0.548 (0.180–1.674) |
| Healing time | 7.65 ± 1.724 | 7.93 ± 1.724 | 0.169 | 0.212 (-0.125–0.709) |

OR:Odds Ratio.

used compared with non-coated suture [1]. After clean contaminated surgery, use of TCS coated suture was associated with 31% reduction of SSI rate [13]. Leaper et al, showed that the effectiveness of TCS coated suture was not affected by the wound type and concluded to a clear benefit of reduction of SSIs after all classes of surgery with the TCS suture [14].

The rate of SSI in our current study was 5.3% showing that this complication remains a common and unsolved issue. A significant difference between TCS coated suture arm and the non-coated suture arm was noted. The use of TCS was associated with a reduction of 70% of SSI rate after adjusting confounding factors (ORa = 0.294 (0.094–0.921); p = 0.036), confirming the efficacy of TCS in reducing the risk of SSI.

Effect of triclosan use in wound healing had been evaluated in several studies. Rasic et al found that TCS suture was associated with less inflammatory reaction, dehiscence and incisional hernia after colorectal surgery [11]. Metavelli et al demonstrated that overall wound complication rate was similar between coated and non-coated suture and they found that TCS coated suture was associated with lower incisional hematoma [5]. After saphenectomy, lower wound pain and wound hyperthermia were associated with TCS suture use [15]. In contrast, other studies failed to demonstrate the superiority of coated suture in reducing wound complications. In a RCT evaluating TCS use after pediatric surgery, dehiscence was similar with coated and non-coated suture [16]. Arselan et al did not found differences in rate of wound dehiscence and demonstrated that the coted suture was associated with a higher risk of wound discharge [17]. In all above studies, wound healing time was not affected by TCS suture use.

In our study, the use of Vicryl® Plus was associated with lower risk of wound oedema, wound hematoma and wound dehiscence. Wound healing time was similar in the two groups. Considering these results, and previous studies, it can be concluded that triclosan coated suture presents an opportunity to improve the postoperative wound healing process.

The present study has notable strengths. To our knowledge this study is the first RCT evaluating TCS coated suture in obstetric surgery. Furthermore, as it was unicentric, potential variables such as surgeon experience, SSI prevention measures, and differences across operating rooms were the same in all time periods.

This study is associated with limitations. The fact that the survey was carried out in a single public institution without extending to other health structures, constitutes one of the limitations of our study. Another weakness is that although the SSI rate in the control group was estimated accurate, detection of absolute reduction of 7% was rather ambitious. In fact, the definition of minimal clinically relevant reduction was extensively debated in our study group at the time of protocol development. Eventually we opted for absolute reduction of 7%, relative reduction of 78%, in the sense that every surgeon will support that 7% absolute risk is clinically meaningful.

## Conclusion

Through this prospective randomized controlled trial, we were able to establish the effectiveness of Triclosan-based antibacterial suture in reducing the incidence of postoperative infections after caesarean section. The integration of TCS coated suture in daily surgical practice results in a significant reduction in overall healthcare costs.

## Supporting information

**S1 Dataset. 10.6084/m9.figshare.21330624.**
(DOCX)

**S1 File. Consort checklist.**
(DOC)

**S2 File. Informed consent form.**
(DOC)

**S3 File. Study protocol.**
(DOCX)

## Author Contributions

**Conceptualization:** Wael Mbarki, Hajer Bettaieb.

**Data curation:** Wael Mbarki, Idriss Laabidi, Hadhemi Rouis, Soumaya Halouani, Rami Boufarghine, Rahma Bouhmida, Mariem Ouederni, Amal Chermiti, Hadir Laamiri.

**Formal analysis:** Wael Mbarki.

**Funding acquisition:** Wael Mbarki.

**Methodology:** Wael Mbarki, Hajer Bettaieb.

**Project administration:** Hajer Bettaieb, Hedhili Oueslati, Chaouki Mbarki.

**Resources:** Amira Lika, Imen Chaibi.

**Software:** Amira Lika, Imen Chaibi, Najeh Hsayaoui.

**Supervision:** Hajer Bettaieb, Hedhili Oueslati, Najeh Hsayaoui, Chaouki Mbarki.

**Validation:** Hajer Bettaieb, Maha Bouyahia, Anissa Ben Amor, Najeh Hsayaoui.

**Writing – original draft:** Wael Mbarki.

**Writing – review & editing:** Hajer Bettaieb, Nesrine Souayeh, Chaouki Mbarki.

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
