## [Decision Letter · Decision Letter 0]

28 Aug 2022

PONE-D-22-20609Evaluation of Triclosan Coated Suture in Obstetrical Surgery: A Prospective Randomized Controlled StudyPLOS ONE

Dear Dr. BETTAIEB,

Thank you for submitting your manuscript to PLOS ONE. After careful consideration, we feel that it has merit but does not fully meet PLOS ONE’s publication criteria as it currently stands. Therefore, we invite you to submit a revised version of the manuscript that addresses the points raised during the review process.

We look forward to receiving your revised manuscript.

Kind regards,

Antonio Simone Laganà, M.D., Ph.D.

Academic Editor

PLOS ONE

Journal Requirements:

2. Please ensure that you have specified (1) whether consent was informed and (2) what type you obtained (for instance, written or verbal, and if verbal, how it was documented and witnessed). If your study included minors, state whether you obtained consent from parents or guardians. If the need for consent was waived by the ethics committee, please include this information.

3. We note that you have selected “Clinical Trial” as your article type. PLOS ONE requires that all clinical trials are registered in an appropriate registry (the WHO list of approved registries is at      https://www.who.int/clinical-trials-registry-platform/network/primary-registries"" https://www.who.int/clinical-trials-registry-platform/network/primary-registries and more information on trial registration is at http://www.icmje.org/about-icmje/faqs/clinical-trials-registration/).

Please state the name of the registry and the registration number (e.g. ISRCTN or ClinicalTrials.gov) in the submission data and on the title page of your manuscript.

a) Please provide the complete date range for participant recruitment and follow-up in the methods section of your manuscript.

b) If you have not yet registered your trial in an appropriate registry, we now require you to do so and will need confirmation of the trial registry number before we can pass your paper to the next stage of review. Please include in the Methods section of your paper your reasons for not registering this study before enrolment of participants started. Please confirm that all related trials are registered by stating: “The authors confirm that all ongoing and related trials for this drug/intervention are registered”.

Please see http://journals.plos.org/plosone/s/submission-guidelines#loc-clinical-trials for our policies on clinical trials.

5. We note you have included a table to which you do not refer in the text of your manuscript. Please ensure that you refer to Table 2 and 3 in your text; if accepted, production will need this reference to link the reader to the Table.

Additional Editor Comments:

The topic of the manuscript is interesting. Nevertheless, the reviewers raised several concerns: considering this point, I invite authors to perform the required major revisions.

Reviewers' comments:

Reviewer's Responses to Questions

**Comments to the Author**

1. Is the manuscript technically sound, and do the data support the conclusions?

Reviewer #1: Yes

Reviewer #2: No

Reviewer #3: Yes

Reviewer #4: Yes

2. Has the statistical analysis been performed appropriately and rigorously? 

Reviewer #1: Yes

Reviewer #2: No

Reviewer #3: Yes

Reviewer #4: No

3. Have the authors made all data underlying the findings in their manuscript fully available?

Reviewer #1: No

Reviewer #2: No

Reviewer #3: Yes

Reviewer #4: Yes

4. Is the manuscript presented in an intelligible fashion and written in standard English?

Reviewer #1: No

Reviewer #2: Yes

Reviewer #3: Yes

Reviewer #4: Yes

5. Review Comments to the Author

Reviewer #1: Dear Authors,

This manuscript tried to assess the effectiveness of Triclosan coated sutures in reducing SSI, the rate of wound healing complications, and the cost impact of triclosan-coated suture use from a hospital perspective. I have reviewed the manuscript and would like to suggest following changes in a section-wise manner.

Overall, there are many grammatical and typographical mistakes.

1. Abstract

Abstract seems well written.

2. Introduction

This section needs to be well written and arranged into relevant paragraphs. The authors should first talk in brief about the SSI, risk factors, basics of the suture in the discussion, and how sutures are related to the development of SSI. Following this, the authors should mention recent studies pertaining to the topic and highlight the literature gap. Then, the authors should state how the current study aims to fill the gap.

3. Methodology

The authors have mentioned inclusion, non-inclusion and exclusion criteria. However, there should be only inclusion and exclusion criteria.

There are grammatical and typographical mistakes. Such as chi2 should be replaced with chi-square, Caesar section with caesarean section, and many more.

Also, format the section in proper paragraphs based on the journal’s guidelines. Please make sure to double-check and address all the mistakes.

4. Results

Please format the tables based on the journal’s guidelines. The statistical values mentioned in brackets are confusing.

5. Discussion

The first paragraph should mention the noteworthy findings of the study. The authors should compare and contrast these findings with other studies in subsequent paragraphs. The discussion is too brief. Also, author should add a paragraph regarding limitations and future implications of this study.

6. Conclusion

The conclusion is well written.

7. References

Please format the references based on the journal’s guidelines.

Reviewer #2: A mono-centric prospective controlled randomized trial in pregnant women with elective or emergency caesarian delivery. Patients, surgeons and outcome’s assessor triple blinded to treatment. Three hundred eighty patients were randomly assigned to closure with polyglactin non coated suture VICRYL, or with polyglactin coated suture VICRYL Plus after caesarian section. The primary outcome was the rate of SSIs within 30 days after surgery. SSI rate was 2.5% in Triclosan group compared to 8.1% with non-coated

suture. Use of Triclosan coated suture was associated with 69% reduction in SSI rate (p = 0.037; OR:0.294; CI95%:0.094 – 0.921). The authors claim a total of 5312 USD (33 USD per procedure) was saved with TCS coated suturing; 590 USD per prevented SSI.

Major comments:

No trial registration given.

It seems to be this trial: https://ichgcp.net/clinical-trials-registry/NCT02847936

However, the sample size in that registered trial is 400 and not 296 patients.

Sample size calculations: expected wound infection rates were 2% with coated suture use versus 9% for the control group. On the basis of a two-sided chi2 test for equality of proportions, 80% power to detect a relative risk reduction of 5%, 296 patients were needed, 148 patients per arm. The sample size is powered to test a null hypothesis of this difference: 9% vs 2% is an ARR of 7% and a RRR of 78%. The actual results showed an ARR of 5.6% (8.1-2.5%) and a RRR of 69%. This means that the null hypothesis may be false as the study is underpowered to correctly test a smaller difference than 7%. In other words, the conclusion of the effectiveness of triclosan coated sutures needs to be toned down considerably, coming from an underpowered study.

What was used as standard skin prep? Specifiy at 'surgical procedure'.

"..a scrub nurse verified the randomization list..." (p. 12) Explain this. How could there be a list that the scrub nurse was allowed to look into. Proper computer randomisation allocates treatment group per patient separately. A pre-randomised list of treatment allocations does not count as proper randomisation.

p.13 who did the wound surveillance and decides whether or not a patient was seen by a senior surgeon for suspicion of wound infection?

Cost estimation section needs to be deleted completely, as well as the related conclusion. This is not a proper nor adequate economic analysis. 'Cesarian cost = Bundle cost + Suture material cost + Hospital stay cost' and 'SSI cost = SSI bundle cost + Drugs cost + Investigation tests cost + Reintervention cost + Hospital stay cost' is not accurate, oversimplified, no details supplied, and only a very rough 'quick & dirty' estimate which must be left out because of imprecision and not following the rules of cost analyses.

Statistical analyses paragraph is too concise, lacks detail.

Table 1 baseline characteristics. Since this is a RCT, no statistics must be performed on baseline characteristics as randomisation should take care of random distribution. Delete p value column and related text in results.

Table 2 risk factor for SSI does not contribute to trial results presentation

p.19 'In multivariate regression, only use of non-coated suture was associated with a higher risk of

SSI.' But none of the multivariate data are shown. Do we simply need to believe the authors without any possibility of peer review of the multivariate data? It's also not made available in supplementary materials.

Reviewer #3: This randomized controlled trial assessed the effectiveness of Triclosan coated suture in reducing SSIs rate after caesarian delivery, and to evaluate the cost impact of antimicrobial suture use. They had interesting finding. And the study is well written.

All information about the procedure was described in detail in Method section.

Only one minor concern.

Abstract: 'CI95%' should be '95%CI'.

Reviewer #4: A randomized controlled study was conducted which aimed to assess the effectiveness of Triclosan coated suture in reducing surgical site infections (SSIs) after cesarean delivery. The secondary aim was to compare rates of wound healing complications. A statistically significant reduction in the SSI rate was observed with the Triclosan coated suture compared to the control. Furthermore, Triclosan was associated with statistically lower risks of wound oedema, dehiscence and hematoma compared to the control.

Major revision

Provide a comprehensive “Statistical analysis” section. List and describe all the statistical methods used for the analysis.

Minor revisions:

1- Abstract: Define the abbreviation SSI.

2- Page 4, last line: Write out chi-square.

3- Page 5, Randomization: Provide further details of the randomization process. If block randomization was used, state the block size.

4- Page 8, Statistical analysis: The statements “p degrees of significance” and “the results were presented with their p” are confusing. Please clarify or restate.

5- Page 8, Statistical analysis: Indicate the type of nonparametric tests used and the specific data that was analyzed with these tests.

6- Page 8: Provide corresponding percentages for the counts in these sentences.

A. “One hundred seventy women were allocated in group 1 and 170 in group 2.”

B. “Ultimately, 160 patients remained in the Triclosan arm and 158 in the control arm . . .”

7-Page 8: Define groups 1 and 2 at first mention.

8- Table I: In addition to means, provide standard deviations. In addition to counts, provide percentages. If the data is not normally distributed, provide median, first and third quartiles.

9- Page 11, Second paragraph: Provide 95% confidence intervals for the 2.5% and 8.1%.

10- In the “Statistical analysis” section indicate that both univariate and multivariate analysis were performed. The univariate results are shown in Table II. Clarify.

11- Page 13, lines 1-3: Clarify which factors remained in the multivariate regression model.

12- In the “Statistical analysis” section list and thoroughly describe the use of all statistical methods. Indicate if univariate or multivariate logistic regression was used for the wound healing disturbances (results shown on page 13 and Table III). For further clarity, in the title of Table III, indicate if the results are univariate or multivariate.

13- Page 13: Provide the discharge rates and summarize the healing times in the two groups.

14- Page 14: Provide measures of dispersion for each hospital cost. Indicate if the summary statistics are means, medians, etc.

15- To assist in the review process, add line numbering to the document.

6. PLOS authors have the option to publish the peer review history of their article (what does this mean?). If published, this will include your full peer review and any attached files.

Reviewer #1: **Yes: **Ayush Anand

Reviewer #2: **Yes: **Prof MA Boermeester

Reviewer #3: **Yes: **Jian-Hong Zhong

Reviewer #4: No

---

## [Author Response · Author response to Decision Letter 0]

13 Oct 2022

Thank you for giving us the opportunity to submit a revised draft of the manuscript “Evaluation of Triclosan Coated Suture in Obstetrical Surgery: A Prospective Randomized Controlled Study”. We appreciate the time and the effort that you and the reviewers dedicated to provide feedback on our manuscript and we are grateful for the insightful comments and valuable improvements to our paper. We have incorporated most of the suggestions made by the reviewers. Those changes are highlighted within the manuscript. Please see below for a point-by-point response to the reviewers’ comments and concerns. All page numbers refer to the revised manuscript file with tracked changes.

Reviewers' Comments to the Authors:

 Reviewer 1: This manuscript tried to assess the effectiveness of Triclosan coated sutures in reducing SSI, the rate of wound healing complications, and the cost impact of triclosan-coated suture use from a hospital perspective. I have reviewed the manuscript and would like to suggest following changes in a section-wise manner. Overall, there are many grammatical and typographical mistakes.

 Comment from reviewer one concerning the Abstract: Abstract seems well written.

 Author response: Thank you.

 Comment from reviewer 1 concerning specific change in the introduction: This section needs to be well written and arranged into relevant paragraphs. The authors should first talk in brief about the SSI, risk factors, basics of the suture in the discussion, and how sutures are related to the development of SSI. Following this, the authors should mention recent studies pertaining to the topic and highlight the literature gap. Then, the authors should state how the current study aims to fill the gap.

 Author response: We agree with the reviewer’s assessment. Accordingly, we have revised the introduction. We included SSI risk factors in page 2 (line 50 to 54), basics of suture and relation between suture and SSI in page 2 (line 55 to 60). Recent studies were cited in page 2 (line 60 to 65). We tried to explain gap in literature in page 3 (line 66 to 70).

 Comment from reviewer 1 concerning the methodology: The authors have mentioned inclusion, non-inclusion and exclusion criteria. However, there should be only inclusion and exclusion criteria. There are grammatical and typographical mistakes. Such as chi2 should be replaced with chi-square, Caesar section with caesarean section, and many more. Also, format the section in proper paragraphs based on the journal’s guidelines. Please make sure to double-check and address all the mistakes. 

Author response: Thank you for pointing this out. Section format was modified. We removed the non-inclusion criteria and we tried to correct all grammatical mistakes by a double-check (page 4 to page 8).

 Comment from reviewer 1 concerning the section “Results”: Please format the tables based on the journal’s guidelines. The statistical values mentioned in brackets are confusing.

Author response: Thank you for pointing this out. All tables format has been modified based on journal’s guidelines.

 Comment from reviewer 1 concerning the section “Discussion”: The first paragraph should mention the noteworthy findings of the study. The authors should compare and contrast these findings with other studies in subsequent paragraphs. The discussion is too brief. Also, author should add a paragraph regarding limitations and future implications of this study.

Author response : We tried to mention our finding in the first paragraph of the discussion in page 13 (line 239 to 243) and finding were compared as recommended in page 14 (line 253 to 260). A paragraph that mention the limitation of the study were added in page 15 (line 286 to 293).

 Comment from reviewer 1 concerning the section “Conclusion”: The conclusion is well written.

Author response: thank you.

 Comment from reviewer 1 concerning the section “References”: Please format the references based on the journal’s guidelines.

Author response : section references has been modified based on journal’s guidelines.

Reviewer 2: A mono-centric prospective controlled randomized trial in pregnant women with elective or emergency caesarian delivery. Patients, surgeons and outcome’s assessor triple blinded to treatment. Three hundred eighty patients were randomly assigned to closure with polyglactin non coated suture VICRYL, or with polyglactin coated suture VICRYL Plus after caesarian section. The primary outcome was the rate of SSIs within 30 days after surgery. SSI rate was 2.5% in Triclosan group compared to 8.1% with non-coated

suture. Use of Triclosan coated suture was associated with 69% reduction in SSI rate (p = 0.037; OR:0.294; CI95%:0.094 – 0.921). The authors claim a total of 5312 USD (33 USD per procedure) was saved with TCS coated suturing; 590 USD per prevented SSI.

 Comment from reviewer 2 concerning the trial registration: No trial registration given.

It seems to be this trial: https://ichgcp.net/clinical-trials-registry/NCT02847936

However, the sample size in that registered trial is 400 and not 296 patients.

Author response: Thank you for pointing this out. The trial registration was not mentioned in the manuscript. Our trial registration does not correspond to the trial mentioned by the reviewer.

The trial registration figures in page 1 at the title and the section “Ethics statement” at page 4 (line 72) in the revised manuscript.

Comment from the reviewer 2 concerning “Sample size calculations”: expected wound infection rates were 2% with coated suture use versus 9% for the control group. On the basis of a two-sided chi2 test for equality of proportions, 80% power to detect a relative risk reduction of 5%, 296 patients were needed, 148 patients per arm. The sample size is powered to test a null hypothesis of this difference: 9% vs 2% is an ARR of 7% and a RRR of 78%. The actual results showed an ARR of 5.6% (8.1-2.5%) and a RRR of 69%. This means that the null hypothesis may be false as the study is underpowered to correctly test a smaller difference than 7%. In other words, the conclusion of the effectiveness of triclosan coated sutures needs to be toned down considerably, coming from an underpowered study.”

Author response: We agree that this is a potential limitation of the study and we tried to figure this weakness and to explain it in the section “Discussion” at the page 15 (line 288 to 293).

Comment from the reviewer 2 concerning “Skin preparation: What was used as standard skin prep? Specify at 'surgical procedure'.

Author response: The skin preparation used was added in section “Surgical procedure’ at page 6 (Line 113).

Comment from the reviewer 2 concerning “The randomization procedure”: “a scrub nurse verified the randomization list..." (p. 12) Explain this. How could there be a list that the scrub nurse was allowed to look into. Proper computer randomization allocates treatment group per patient separately. A pre-randomized list of treatment allocations does not count as proper randomization.

Author response: We agree that the section randomization lacks details and was not well described. In the revised manuscript we tried to clarify and to meticulously describe the procedure of randomization in the section “Randomization and blinding” at the page 5 (line 95 to 104).

Comment from the reviewer 2 concerning “the wound surveillance”: who did the wound surveillance and decides whether or not a patient was seen by a senior surgeon for suspicion of wound infection?

Author response: Thank you for pointing this out. Wound surveillance was made by the resident in charge on the ward. Page 6 (line 124).

Comment from the reviewer 2 concerning “Cost estimation”: Cost estimation section needs to be deleted completely, as well as the related conclusion. This is not a proper nor adequate economic analysis. 'Cesarian cost = Bundle cost + Suture material cost + Hospital stay cost' and 'SSI cost = SSI bundle cost + Drugs cost + Investigation tests cost + Reintervention cost + Hospital stay cost' is not accurate, oversimplified, no details supplied, and only a very rough 'quick & dirty' estimate which must be left out because of imprecision and not following the rules of cost analyses.

Author response: We agree with the reviewer that the method to estimate cost was not appropriate and the section and its related conclusion were deleted from the revised manuscript.

Comment from the reviewer 2 concerning the section “Statistical analysis”: Statistical analyses paragraph is too concise, lacks detail.

Author response: We tried to clarify the paragraph by listing and describing all the statistical methods used for the analysis at page 7 and 8 (line 154 to 167).

Comment from the reviewer 2 concerning Table 1: Table 1 baseline characteristics. Since this is a RCT, no statistics must be performed on baseline characteristics as randomization should take care of random distribution. Delete p value column and related text in results.

Author response: We totally agree with the reviewer and p value column and related text was deleted.

Comment from the reviewer 2 concerning Table 2: Table 2 risk factor for SSI does not contribute to trial results presentation.

Author response : While we appreciate the reviewer’s feedback, we respectfully disagree. Since development of SSI is multifactorial, the bivariate analysis was conducted to define confounding factors.

Comment from the reviewer 2 concerning “Multivariate analysis”: p.19 'In multivariate regression, only use of non-coated suture was associated with a higher risk of SSI.' But none of the multivariate data are shown. Do we simply need to believe the authors without any possibility of peer review of the multivariate data? It's also not made available in supplementary materials.

Author response: As suggested results of multivariate analysis were added in the revised manuscript at page 12. 

Reviewer 3: This randomized controlled trial assessed the effectiveness of Triclosan coated suture in reducing SSIs rate after caesarian delivery, and to evaluate the cost impact of antimicrobial suture use. They had interesting finding. And the study is well written. All information about the procedure was described in detail in Method section. Only one minor concern 'CI95%' should be '95%CI'.

Author response: Thank you.

Reviewer 3: A randomized controlled study was conducted which aimed to assess the effectiveness of Triclosan coated suture in reducing surgical site infections (SSIs) after cesarean delivery. The secondary aim was to compare rates of wound healing complications. A statistically significant reduction in the SSI rate was observed with the Triclosan coated suture compared to the control. Furthermore, Triclosan was associated with statistically lower risks of wound oedema, dehiscence and hematoma compared to the control.

Comment from the reviewer 3 concerning the section “Statistical analysis”:

 Provide a comprehensive “Statistical analysis” section. List and describe all the statistical methods used for the analysis.

 Page 4, last line: Write out chi-square

 The statements “p degrees of significance” and “the results were presented with their p” are confusing. Please clarify or restate.

 Indicate the type of nonparametric tests used and the specific data that was analyzed with these tests..

Author response: We tried to clarify the paragraph by Listing and describing all the statistical methods used for the analysis at page 7 and 8 (line 154 to 167).

Comment from the reviewer 3 concerning the section”Abstact”: Define the abbreviation SSI.

Author response : Thank you for pointing this out. The abbreviation has been defined.

Comment from the reviewer 3 concerning the section “Statistical analysis”: 

 Page 4, last line: Write out chi-square

 The statements “p degrees of significance” and “the results were presented with their p” are confusing. Please clarify or restate.

 Indicate the type of nonparametric tests used and the specific data that was analyzed with these tests.

Comment from the reviewer 3 concerning “Providing percentage”: Provide corresponding percentages for the counts in these sentences.

A. “One hundred seventy women were allocated in group 1 and 170 in group 2.”

B. “Ultimately, 160 patients remained in the Triclosan arm and 158 in the control arm

Author response: Thank you for pointing this out. Percentage has been added at page 8 (line 169 to 174).

Comment from the reviewer 3 concerning: Define groups 1 and 2 at first mention.

Author response: Thank you for pointing this out. The two groups were defined as recommended at page 5 (Line 93 and 94).

Comment from the reviewer 3 concerning” Table 1”: In addition to means, provide standard deviations. In addition to counts, provide percentages. If the data is not normally distributed, provide median, first and third quartiles.

Author response: Standard deviation, percentage were added in table 1 to normally distributed data and median, first and third quartiles were also mentioned. Page 9

Comment from the reviewer 3 concerning “Adding standard deviation”: Page 11, Second paragraph: Provide 95% confidence intervals for the 2.5% and 8.1%.

Author response: These two percentages define the rate of SSI in the two groups. No standard deviation could be calculated.

Comment from the reviewer 3 concerning “Multivariate analysis”:

 In the “Statistical analysis” section indicate that both univariate and multivariate analysis were performed. The univariate results are shown in Table II. Clarify.

 Clarify which factors remained in the multivariate regression.

Author response: As suggested results of multivariate analysis were added in the revised manuscript at page 12. 

Comment from the reviewer 3 concerning: In the “Statistical analysis” section list and thoroughly describe the use of all statistical methods. Indicate if univariate or multivariate logistic regression was used for the wound healing disturbances (results shown on page 13 and Table III). For further clarity, in the title of Table III, indicate if the results are univariate or multivariate.

Author response: Thank you for pointing this out. In the revised manuscript we had mentioned that univariate analysis was used to compare wound healing disturbances at page 8 line 165.

Comment from the reviewer 3 concerning: Providing the discharge rates and summarizing the healing times in the two groups.

Author response: We described these two parameters at page 12 (line 224 to 226) and in Table 3.

Comment from the reviewer 3 concerning “Hospital cost”: Provide measures of dispersion for each hospital cost. Indicate if the summary statistics are means, medians, etc.

Author response: The section of hospital cost has been deleted as recommended.

Comment from the reviewer 3 concerning “Adding line numbering to the document” : To assist in the review process, add line numbering to the document.

Author response: Thank you it was a very helpful suggestion.

Reviewer 4: A randomized controlled study was conducted which aimed to assess the effectiveness of Triclosan coated suture in reducing surgical site infections (SSIs) after cesarean delivery. The secondary aim was to compare rates of wound healing complications. A statistically significant reduction in the SSI rate was observed with the Triclosan coated suture compared to the control. Furthermore, Triclosan was associated with statistically lower risks of wound oedema, dehiscence and hematoma compared to the control.

Major revision

Provide a comprehensive “Statistical analysis” section. List and describe all the statistical methods used for the analysis.

Author response: the statistical analysis section was revised. We tried to clarify the paragraph by listing and describing all the statistical methods used for the analysis at page 7 and 8 (line 154 to 167).

Minor revisions:

1- Abstract: Define the abbreviation SSI.

2- Page 4, last line: Write out chi-square.

3- Page 5, Randomization: Provide further details of the randomization process. If block randomization was used, state the block size.

4- Page 8, Statistical analysis: The statements “p degrees of significance” and “the results were presented with their p” are confusing. Please clarify or restate.

5- Page 8, Statistical analysis: Indicate the type of nonparametric tests used and the specific data that was analyzed with these tests.

6- Page 8: Provide corresponding percentages for the counts in these sentences.

A. “One hundred seventy women were allocated in group 1 and 170 in group 2.”

B. “Ultimately, 160 patients remained in the Triclosan arm and 158 in the control arm . . .”

7-Page 8: Define groups 1 and 2 at first mention.

8- Table I: In addition to means, provide standard deviations. In addition to counts, provide percentages. If the data is not normally distributed, provide median, first and third quartiles.

9- Page 11, Second paragraph: Provide 95% confidence intervals for the 2.5% and 8.1%.

10- In the “Statistical analysis” section indicate that both univariate and multivariate analysis were performed. The univariate results are shown in Table II. Clarify.

11- Page 13, lines 1-3: Clarify which factors remained in the multivariate regression model.

12- In the “Statistical analysis” section list and thoroughly describe the use of all statistical methods. Indicate if univariate or multivariate logistic regression was used for the wound healing disturbances (results shown on page 13 and Table III). For further clarity, in the title of Table III, indicate if the results are univariate or multivariate.

13- Page 13: Provide the discharge rates and summarize the healing times in the two groups.

14- Page 14: Provide measures of dispersion for each hospital cost. Indicate if the summary statistics are means, medians, etc.

15- To assist in the review process, add line numbering to the document.

Thank you for appointing this out. All minor revision were considered and clarified as recommended by reviewer n°4.

---

## [Decision Letter · Decision Letter 1]

26 Oct 2022

PONE-D-22-20609R1Evaluation of Triclosan Coated Suture in Obstetrical Surgery: A Prospective Randomized Controlled StudyPLOS ONE

Dear Dr. BETTAIEB,

Thank you for submitting your manuscript to PLOS ONE. After careful consideration, we feel that it has merit but does not fully meet PLOS ONE’s publication criteria as it currently stands. Therefore, we invite you to submit a revised version of the manuscript that addresses the points raised during the review process.

We look forward to receiving your revised manuscript.

Kind regards,

Antonio Simone Laganà, M.D., Ph.D.

Academic Editor

PLOS ONE

Journal Requirements:

Additional Editor Comments:

The reviewers have still some concerns that need to be addressed by the authors.

Please check carefully the English grammar and style.

Reviewers' comments:

Reviewer's Responses to Questions

**Comments to the Author**

1. If the authors have adequately addressed your comments raised in a previous round of review and you feel that this manuscript is now acceptable for publication, you may indicate that here to bypass the “Comments to the Author” section, enter your conflict of interest statement in the “Confidential to Editor” section, and submit your "Accept" recommendation.

Reviewer #1: (No Response)

Reviewer #3: All comments have been addressed

Reviewer #4: (No Response)

2. Is the manuscript technically sound, and do the data support the conclusions?

Reviewer #1: Partly

Reviewer #3: Yes

Reviewer #4: Yes

3. Has the statistical analysis been performed appropriately and rigorously? 

Reviewer #1: Yes

Reviewer #3: Yes

Reviewer #4: Yes

4. Have the authors made all data underlying the findings in their manuscript fully available?

Reviewer #1: Yes

Reviewer #3: Yes

Reviewer #4: (No Response)

5. Is the manuscript presented in an intelligible fashion and written in standard English?

Reviewer #1: No

Reviewer #3: Yes

Reviewer #4: Yes

6. Review Comments to the Author

Reviewer #1: Dear Author, I have reviewed the manuscript and unfortunately, I will not be able to recommend it for publication. This manuscript needs to be extensively edited for english language. Discussion section is not written well.

Reviewer #3: All my comments have been addressed. I think the paper is suitable for publication in PLOS ONE. Thanks again.

Reviewer #4: Minor Revisions:

1- Lines 159 & 165: State the statistical method(s) used for the bivariate and univariate analyses.

2- Table I: To improve clarity, place "n=160" in the Group 1 header and "n=158" in the Group 2 header. Remove these numbers from the cell entries.

3- Lines 190 and 195: Provide 95% confidence intervals for the proportions: 5.3%, 2.5%, and 8.1%. In the Statistical Analysis section, include the statistical method used to estimate the CIs.

4- Table II:

a- In the title state the statistical method used for the bivariate analysis.

b- Provide a measure of dispersion for all means.

c- In similar fashion to comment #2 above, include the sample sizes in the header row and remove them from the individual cells.

5- Table IV: In similar fashion to comment #2 above, include the sample sizes in the header row and remove them from the individual cells.

7. PLOS authors have the option to publish the peer review history of their article (what does this mean?). If published, this will include your full peer review and any attached files.

Reviewer #1: **Yes: **Ayush Anand

Reviewer #3: **Yes: **Jian-Hong Zhong

Reviewer #4: No

---

## [Author Response · Author response to Decision Letter 1]

28 Oct 2022

Thank you for giving us the opportunity to submit a revised draft of the manuscript “Evaluation of Triclosan Coated Suture in Obstetrical Surgery: A Prospective Randomized Controlled Study”. We appreciate the time and effort that you and the reviewers dedicated to provide feedback on our manuscript and we are grateful for the insightful comments on and valuable improvements to our paper. We have incorporated most of the suggestions made by the reviewers. Those changes are highlighted within the manuscript. Please see below, in blue, for a point-by-point response to the reviewers’ comments and concerns. All page numbers refer to the revised manuscript file with tracked changes.

Reviewers' Comments to the Authors:

 Reviewer 1 : Dear Author, I have reviewed the manuscript and unfortunately, I will not be able to recommend it for publication. This manuscript needs to be extensively edited for english language. Discussion section is not written well.

 Author response : We tried to write the manuscript in a clear language and to correct grammatical and typographical errors.

Reviewer 3 : All my comments have been addressed. I think the paper is suitable for publication in PLOS ONE. Thanks again

Author response : Thank you.

Reviewer 4: 

 Comment from reviewer 4 concerning bivariate analysis “Lines 159 & 165: State the statistical method(s) used for the bivariate and univariate analyses. “

 Author response: Thank you for pointing this out. We tried to describe all statistical methods used for the bivariate analysis in the page 7 and 8 (line 156 to line 159).

Comment from reviewer 4 concerning Table I, Table II and Table IV:

 To improve clarity, place "n=160" in the Group 1 header and "n=158" in the Group 2 header. Remove these numbers from the cell entries.

 Table II:

 a- In the title state the statistical method used for the bivariate analysis.

 b- Provide a measure of dispersion for all means.

 c- In similar fashion to comment #2 above, include the sample sizes in the header row and remove them from the individual cells.

Table IV: In similar fashion to comment #2 above, include the sample sizes in the header row and remove them from the individual cells.

Author response: We modified Tables as recommended.

Comment from reviewer 4 concerning Providing 95% confidence intervals for the proportions “Lines 190 and 195: Provide 95% confidence intervals for the proportions: 5.3%, 2.5%, and 8.1%. In the Statistical Analysis section, include the statistical method used to estimate the Cis.

Author response: Thank you for the reminder. 95%Cis were provided as recommended and method used was included in page 8 (line 165 to line 166).

---

## [Editor Report · Decision Letter 2]

25 Nov 2022

Evaluation of Triclosan Coated Suture in Obstetrical Surgery: A Prospective Randomized Controlled Study

PONE-D-22-20609R2

Dear Dr. BETTAIEB,

We’re pleased to inform you that your manuscript has been judged scientifically suitable for publication and will be formally accepted for publication once it meets all outstanding technical requirements.

Kind regards,

Antonio Simone Laganà, M.D., Ph.D.

Academic Editor

PLOS ONE

Additional Editor Comments (optional):

I carefully evaluated the revised version of this manuscript.

Authors have performed the required changes, improving significantly the quality of the paper.
---

## [Editor Report · Acceptance letter]

2 Dec 2022

PONE-D-22-20609R2 

Evaluation of Triclosan Coated Suture in Obstetrical Surgery: A Prospective Randomized Controlled Study (NCT05330650). 

Dear Dr. BETTAIEB:

I'm pleased to inform you that your manuscript has been deemed suitable for publication in PLOS ONE. Congratulations! Your manuscript is now with our production department. 

Kind regards, 

on behalf of

Dr. Antonio Simone Laganà 

Academic Editor

PLOS ONE